# Violence against women migrant workers in Thailand: A cross-sectional study on experiences, impact, and help seeking

Montakarn Chuemchit[1]☯*, Nyan Linn[2]☯, Chit Pyae Pyae Han[2], Zayar Lynn[2], Suttharuethai Chernkwanma[2], Nutta Taneepanichskul[2], Wandee Sirichokchatchawan[2], Ratana Somrongthong[2]

1 College of Public Health Sciences, Chulalongkorn University and Excellent Center for Health and Social Sciences and Addiction Research, Bangkok, Thailand, 2 College of Public Health Sciences Chulalongkorn University, Bangkok, Thailand

☯ These authors contributed equally to this work as the first authors.

* Montakarn.Ch@chula.ac.th

## Abstract

### Background

Violence against women migrant workers is an alarming public health problem. This study aims to investigate the experiences and impact of violence on women migrant workers in Thailand, including their help-seeking patterns, knowledge, future plans for seeking help, ways of help received, and the factors associated with violence.

### Methods

This cross-sectional study included 494 women migrant workers across seven key provinces in Thailand – Bangkok, Chiang Rai, Chonburi, Samut Prakan, Rayong, Samut Sakhon, and Tak. Data was collected by well-trained enumerators using the validated standardized structured questionnaire. We analysed multiple logistic regressions to investigate the factors associated with experiences of violence and adjusted odds ratios (AOR) with a 95% CI were presented.

### Results

The study identified the lifetime prevalence of violence to be 54.45%, 42.47% for intimate partner violence (IPV), and 46.15% for non-intimate partner violence (N-IPV). Women experienced psychological, physical, and occupational impacts from violence. A significant proportion of women sought informal support from friends (59.1%) and family (52.6%), while formal support was most commonly sought from employers (15.3%) and the police (6.6%). Factors associated with IPV and/or N-IPV among women migrant workers included area of residence, duration of stay, monthly

**Data availability statement:** All relevant data are within the manuscript and its Supporting Information files. Quantitative data is openly accessible.We have uploaded the dataset to the system. The qualitative data are not presented/ described/used/ analysed in this paper. Therefore, the qualitative data is not relevant to mention for this current paper.

**Funding:** This study was funded by the EU funded Spotlight initiative, Safe and Fair Program, UN Women Thailand (PSA-ROAP-2021-008). The funders had no role in study design, data collection and analysis, decision to publish, or preparation of the manuscript.

**Competing interests:** The authors have declared that no competing interests exist.

income, perceived safety, Thai language proficiency, and job or income loss during the COVID-19 pandemic.

## Conclusion

The findings highlight widespread violence against migrant women workers in Thailand, urging the need for tailored support services. Key strategies include strengthening workplace-based protections, multilingual reporting mechanisms, peer networks, culturally relevant education, and individual- and community-level interventions within labour, health, and social protection systems. Collaborative efforts among policymakers and stakeholders are vital for the prevention of and response to violence and for ensuring the safety of women migrant workers.

---

## Introduction

A migrant worker is "a person who migrates from one country to another intending to be employed otherwise than on his or her own account" [1]. There were about 169 million migrant workers internationally in 2019, of which about 41.5% were women [2]. Thailand is a developing country in Southeast Asia. There were approximately 2.5 million documented migrant workers in Thailand, with an estimated 53% from Myanmar, 24% from Cambodia, and 22% from the Lao People's Democratic Republic (Lao PDR) [3]. In addition, there are an estimated two million undocumented migrant workers in Thailand [4].

Thailand has been a destination for migrant workers from neighbouring countries in the past three decades, and they have played a significant role in the development of Thailand's economy. Migrant workers contributed more than 6% of the country's Gross Domestic Product in 2010 and occupied nearly 5% of the jobs [5]. Most migrant workers in Thailand are engaged in low-paying occupations, such as domestic work, construction, manufacturing, fishing, and agriculture. Women constitute about half of all migrant workers in Thailand [3], and their migration is often related to the difficult socioeconomic situations of their families in their home countries [6], Being women and migrants, they are susceptible to violence [7].

The United Nations (UN) defines violence against women (VAW) as "any act of gender-based violence that results in or is likely to result in, physical, sexual, or mental harm or suffering to women, including threats of such acts, coercion or arbitrary deprivation of liberty, whether occurring in public or in private life" [8]. The two major types of VAW are intimate partner violence (IPV) and non-intimate partner violence (N-IPV) [9]. In addition to violence, women migrant workers in Thailand frequently experience domestic abuse, discrimination, and abusive labour practices such as low pay, long working hours, and inadequate social security [10,11].

Violence against women migrant workers is a globally significant public health problem. The UN estimates that one in three women have been subjected to at least one form of violence in their lifetime [12]. VAW can lead to serious physical health problems, including physical injury, chronic pain, and reproductive health problems,

especially sexually transmitted infections (STIs). Moreover, VAW can lead to mental health problems, such as depression, anxiety and post-traumatic stress disorder (PTSD). In addition, VAW incurs profound social and economic losses for survivors, their families, and society as a whole [13].

Moreover, VAW is a serious violation of human rights, including "the right to life", "the right not to be tortured or treated in an inhuman and degrading way", "the right to respect for private and family life including the right to physical and psychological integrity", and "the right not to be discriminated against" [8]. Accordingly, the UN made the "Declaration on the Elimination of Violence Against Women" in 1993, which addressed VAW and presented an action framework for the global and national levels [8]. In addition, Sustainable Development Goal 5 by the UN aims to achieve gender equality and empower all women and girls with the targets of ending discrimination and violence against and the exploitation of women and girls [14]. However, women migrant workers in Thailand are frequently subjected to violence and have difficulties accessing legal protections and labour benefits [15,16].

The COVID-19 pandemic has brought unprecedented challenges, which have disproportionately affected vulnerable populations, particularly female migrant workers in Thailand. The combination of pandemic-induced uncertainty associated with lockdown measures and deep-seated inequalities has magnified these women's vulnerabilities. Understanding the specific dynamics of violence against women migrant workers during this time is imperative. The pandemic not only exposed these women to a heightened risk of violence and discrimination; it also engendered an environment of silence, making reporting such abuse difficult [15,17,18]. The economic downturn caused by the pandemic resulted in a chain reaction of negative consequences, including job losses and loss of legal status, making these women even more vulnerable to exploitation. Language barriers, fear of punishment, and social isolation exacerbated by the pandemic's ripple effects further marginalized them, limiting their access to critical services such as healthcare and legal assistance [19,20]. These challenges not only increase the risk of existing violence but may also give rise to new forms of mistreatment. This context may increase women's exposure to existing violence by intimate partners and non-partners, and new forms of VAW may emerge.

Accordingly, it is crucial to study the women migrant workers' experiences of violence during the COVID-19 pandemic in Thailand. However, the number and scope of studies on this topic are inadequate, and there remains a research gap, particularly in terms of a detailed understanding of violence experiences among women migrants [21,22]. Therefore, this study aimed to investigate the experiences and impact of violence, help-seeking patterns, knowledge, and future plan of help seeking, and ways of help received among women migrant workers in Thailand. This study will contribute important findings for the development of policies and programs to address VAW among women migrant workers in Thailand.

## Materials and methods

This study is part of a broader research project on violence against women migrant workers in Thailand. Partial findings were previously published in (Chuemchit et al., 2024) [23], focusing on discrimination during the COVID-19 pandemic and its associated factors. Although this study uses the same sample from the research project and therefore shares general characteristics with the preceding paper, the analyses and results presented in the current paper, including all tables and figures, are entirely novel. This paper presents new analyses and critical findings on women's lifetime experiences of IPV and non-IPV, their impacts, associated factors, and patterns of help-seeking, including ways help was received, knowledge of help-seeking, and future help-seeking plans. Accordingly, the current paper addresses separate analytical aims and contributes to original evidence that is not reported in the preceding publication and therefore does not constitute redundant publication. Findings of this paper provide a significant contribution to understanding the multifaceted challenges faced by migrant women workers.

### Study design and sampling techniques

This cross-sectional study was conducted across key provinces, Bangkok, Chiang Rai, Chonburi, Samut Prakan, Rayong, Samut Sakhon, and Tak. We employed a snowball sampling technique as the study population is hard to reach. The

inclusion criteria include 1) being women migrant workers, whether either documented or undocumented; 2) being aged 18 years or above; 3) having resided in Thailand for at least one month; 3) having as their country-of-origin Lao PDR, Myanmar, or Cambodia; and 4) being able to communicate in their mother tongue.

The sample size was determined by feasibility considerations and recruitment capacity, given the hard-to-reach nature of migrant women workers and the use of snowball sampling with a total sample of 494 participants, the study had adequate precision to estimate prevalence for common outcomes and sufficient power to detect associations of moderate magnitude in logistic regression analyses. Assuming a two-sided alpha level of 0.05 and outcome prevalence between 20% and 50%, this sample size would allow detection of odds ratios in the moderate range for exposures with reasonable prevalence in the sample. No formal a priori sample size or power calculation was conducted; therefore, smaller effect sizes or associations involving rare exposures may not have been detectable, and nonsignificant findings should be interpreted with caution.

Initial seeds were purposively selected through community-based organizations and outreach workers based on eligibility, willingness to participate, and diversity in nationality, age, and employment sector. To enhance diversity in recruitment chains, multiple seeds were initiated across different geographic locations, workplace settings, and migrant communities. Recruitment was proceeded through participant referral chains, with each participant invited to refer additional eligible participants from their social networks. Recruitment continued for approximately five to ten referral waves, until the target sample size was achieved, and no substantial new recruitment networks emerged. The demographic characteristics of the sample, including age, nationality, and employment sector, were compared with available national and regional statistics on migrant workers in Thailand to assess the degree of representativeness.

## Research instruments

The questionnaire was developed in the English, Khmer, Lao, and Burmese languages, and its content validity was checked and validated by three experts. Participant's Information included women's age, marital status, educational background, and duration of residence in Thailand. Questions regarding the impacts of COVID-19 Pandemic covered areas such as employment status, financial stability, and access to healthcare during the pandemic.

General Safety (adapted from the WHO multi-country study on women's health and domestic violence [24]: This section assessed women's experiences of IPV and N-IPV, as described in Table 1. For each questionnaire item of violence, women were asked about the frequency (once, a few times, or many times) and time (during the COVID-19 period [within the last 12 months], before the last 12 months [lifetime], or both). The reliability coefficient of this tool was 0.8 for IPV and 0.9 for N-IPV in this study.

The survey instrument was adapted with additional items developed to capture emerging forms of violence, including economic and cyber violence. Content validity was assessed by three purposively selected experts with expertise in gender-based violence, migrant health, and public health research, who reviewed the instrument for relevance, clarity, cultural appropriateness, and construct coverage. The questionnaire was translated using a forward–backward translation process conducted by bilingual translators, with discrepancies resolved by the research team to ensure conceptual equivalence. The instrument was pretested with participants from the target population to assess clarity, comprehension, and flow, and minor revisions were made accordingly.

Impact of violence section asked about the emotional, physical, and socio-economic consequences. Coping and knowledge section investigated survivors' coping mechanisms in response to violence, assessed their knowledge of available support services, and future plans of help-seeking.

## Data collection

This study was approved by the Chulalongkorn University Ethics Review Committee (COA.No.232/2020), Thailand, on 8 October 2020, with an approved extension under COA.No.196/2021 on 8 October 2021. Each participant provided written

**Table 1. Questionnaire items used to ask about women's experience of lifetime violence.**

| Economic violence | Cyber violence | Psychological violence | Physical violence | Sexual violence |
|---|---|---|---|---|
| Restricted from going to work, trading, making money, or making plans to earn income | Insulted or harassed by text message, social media message, or phone call | Insulted; made to feel bad about herself; something to embarrass her said in front of others | Slapped on the cheek; had something that could injure her thrown at her; pushed; locked up; hair forcefully pulled | Forced to have sex when she did not want to; Was threatened or coerced |
| Income forcibly taken against her will | Threatened via text message, email, or social media | Things that are important to her destroyed; scared; manipulated | Punched; beaten to hurt her; dragged | Had undesired sexual intercourse because she was afraid of what might happen if she refused |
| Evicted from home | | Threatened to harm her or someone she cares about | Strangulated; intentionally burnt | Forced to do any sexual acts that she did not want to, that affected her dignity, which were disgusting, or that were shameful |
| | | | Threatened or harmed with a knife or other weapon | |

and verbal informed consent. The participant could waive the written consent by not signing the consent form, their participation was completely voluntary, and their identity was kept confidential.

Our research team conducted the data collection from 12/12/2021–27/05/2022. Enumerators conducted the quantitative survey in each province. They were female migrants who had experience doing research among migrants and were well-acquainted with the violence study. Before the data collection, the researchers conducted a comprehensive training of the enumerators, either face-to-face or online, together with partnered organizations. The training consisted of a brief explanation of the study, including its objectives, screening criteria, and interview tools, and detailed how to communicate with participants ethically and in a gender-sensitive way. Through this training, the enumerators were able to develop skills and experiences in gender-based violence research and engage well in the research, which provided them with paid employment.

We approached women migrants in each province through partnered organizations, and they were invited to participate in interviews. After screening each woman's eligibility, the enumerator obtained informed consent from the participant before proceeding with data collection, a face-to-face interview in a setting that ensured privacy and confidentiality. This study did not involve any minor participants; therefore, parental or guardian consent was not required. The participants had autonomy to skip a question, pause, or stop during the interview. After the interview, the enumerator provided each woman with contact information for healthcare services, social support services, the police, and her home's country's embassy. In addition, the enumerator provided urgent referrals to healthcare or social support services to women who required them.

## Statistical analysis

We used the SPSS software version 28.0 (IBM SPSS, version 28, Armonk, NY, USA; Chulalongkorn University license) for data analysis. There was no missing data. In univariate analysis, we described the variables using mean, standard deviation (SD), median, interquartile range (IQR), frequency, and percentage (%). In inferential analyses, two outcomes, lifetime IPV and lifetime N-IPV, were separately analysed as dichotomous variables (0 = never, 1 = ever). For bivariate analysis, we analysed simple logistic regressions and presented Crude odds ratios (COR) and 95% confidence intervals (CI). For multivariable analysis, multiple logistic regressions were analysed adjusting for age, education level, and marital status and adjusted odds ratios (AOR) with a 95% CI were presented. These variables were selected as adjustment variables based on prior literature and conceptual relevance, as they are potentially associated with both exposure variables and violence

outcomes. Other variables such as income and length of stay were conceptually defined as primary exposure variables rather than confounders. Adjusting for these variables could constitute over adjustment and potentially obscure the associations of interest. Multicollinearity among predictors was tested; Variance Inflation Factor (VIF) of all variables were ≤ 5, indicating no issue of multicollinearity. Variables with *p-value* less than 0.05 were considered statistically significant.

## Results

### General characteristics

Among the 494 women migrant workers included in the study, the average age was $27 \pm 13$ years. Most women had received secondary school education level (47.2%), and more than half of women were married or lived in a union with their partner (68.2%). More than 40% of women identified Myanmar as their country of origin (40.9%) and 70.1% of women had stayed in Thailand for a duration of five or more years. The average income (THB per month) in this study was 5,001–10,000 for 33.2% of women and 10,001–15,000 for 39.1% of women (1USD ≈ 33 THB). The occupations of the participants in the last 12 months in Thailand are illustrated in Fig 1.

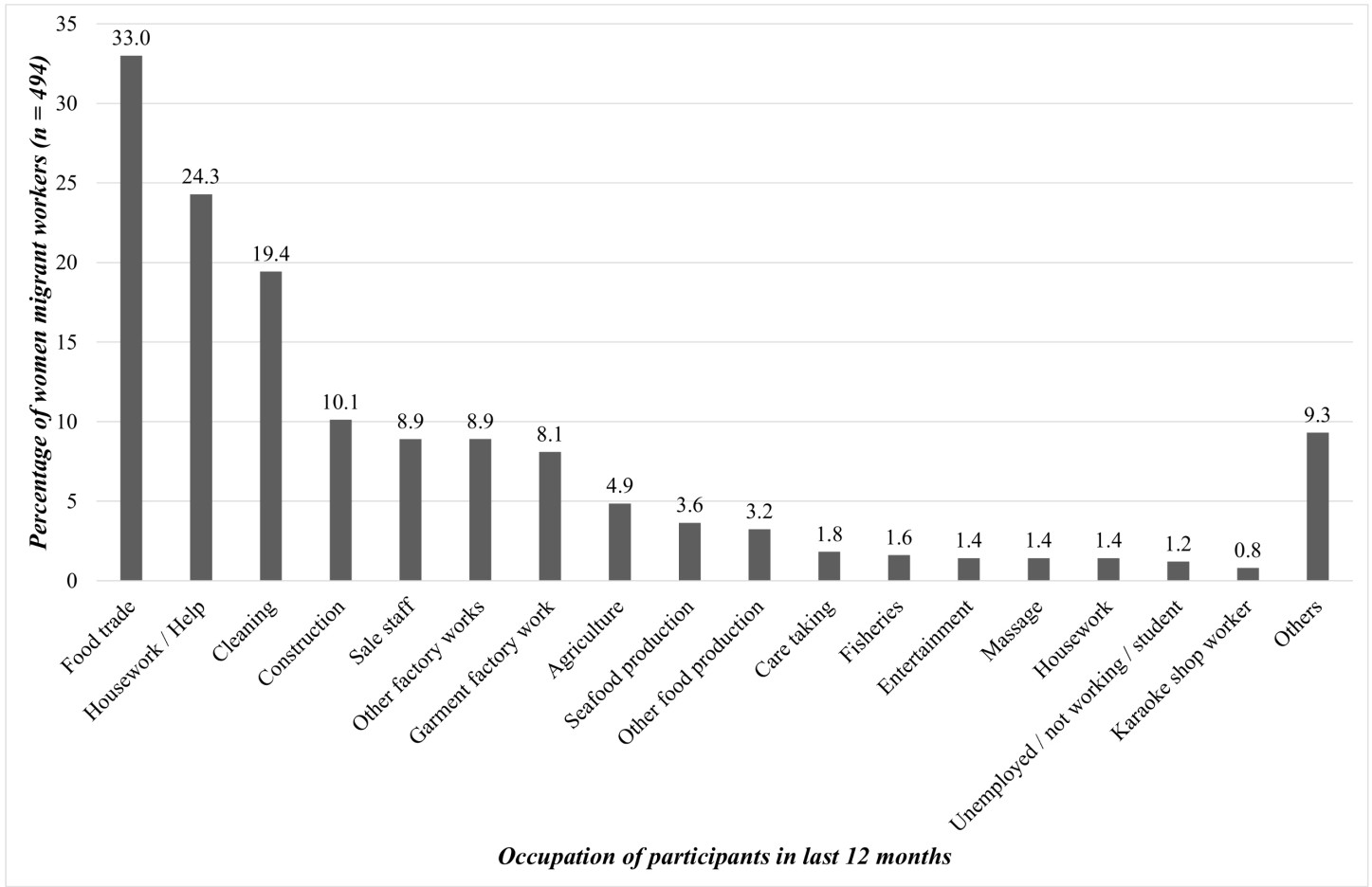

**Fig 1. Occupations of the participants in the last 12 months in Thailand (n = 494).** Note: A participant might work in more than one job. The percentage was calculated separately for each column based on n = 494..

## Experiences of lifetime violence among women migrant workers

As in Table 2, the lifetime prevalence of violence against women migrant workers in this study was 54.45%, with 42.47% for IPV and 46.15% for N-IPV. IPV: Women migrant workers experienced different forms of partner violence to varying degrees. The most commonly reported form of violence was psychological violence (25.93%), followed by economic abuse and financial control (21.73%), physical violence (21.23%), sexual violence (13.83%), and cyberstalking and cyberbullying (11.85%). Non – IPV: The most commonly reported form of violence was psychological violence (26.72%), followed by economic abuse and financial control (19.43%), sexual violence (16.40%), physical violence (10.93%), and cyberstalking and bullying (4.45%).

## Help-seeking of women migrant workers regarding violence

We asked help-seeking of women migrant workers regarding violence by allowing multiple responses to each question. Over half of the participants (n = 273; 55.3%) knew where migrant women workers could seek help if they experienced violence. Formal support, particularly reporting to the police, was highly responded (54.6%) than informal support from family (26.4%). Among the women who actually sought help (n = 137; 50.9% of survivors), informal support was more commonly utilized than formal support, with 59.1% turning to friends and 52.6% to family. Formal support was less frequently sought, with 15.3% reporting to employers, 7.3% to civil society or support services, and 6.6% to the police. These findings show that while many women are aware of formal help sources, they are more likely to rely on informal networks such as friends and family in practice, indicating that knowledge of formal resources does not always translate into action. Regarding future plans for help-seeking among all participants (n = 494), the most mentioned formal supports were the police (30.0%) and employers (29.2%), while the most preferred informal supports were friends (56.7%) and family (23.5%; Table 3).

Regarding the types of help received, we interviewed 137 women survivors who sought help and answers with multiple responses were allowed. The most common ways of receiving help were "stepping in to halt the violence" (27.0%), "being asked about the preferred mode of assistance" (21.9%), and "being given money" (18.3%) (Table 4).

## Impact of violence among survivors

Questions about the impact of violence were asked towards all women survivors of violence (n = 269), and multiple responses were allowed (Table 5). Among women survivors, 50.9% felt scared, insecure, or worried; 42.0% of women felt angry; and 40.9% felt ashamed or embarrassed about the psychological impacts. Regarding physical impacts, 23.8% of women had experienced a physical illness or injury. Regarding impacts on the job, the most common answers were "could not focus on work" (21.6%) and "being distracted while working" (15.2%).

**Table 2. Percentage of lifetime experience of different types of violence.**

| Type of Violence | Intimate Partner Violence (n = 405) | Non-Intimate Partner Violence (n = 494) |
|---|---|---|
| Economic abuse | 21.73 | 19.43 |
| Cyberbullying and social media abuse | 11.85 | 4.45 |
| Psychological violence | 25.93 | 26.72 |
| Physical violence | 21.23 | 10.93 |
| Sexual violence | 13.83 | 16.40 |
| Lifetime prevalence of intimate partner violence (n = 405) = 42.47% | | |
| Lifetime prevalence of non-intimate partner violence (n = 494) = 46.15% | | |
| Lifetime prevalence of IPV and/or N-IPV (n = 494) = 54.45% | | |

**Table 3. Help-seeking patterns, Knowledge, and future plan of help seeking.**

| Supports | | Knowledge of help-seeking (n = 273)# | | Help-seeking patterns (n = 137)^ | | Future plan for help-seeking (n = 494)* | |
|---|---|---|---|---|---|---|---|
| | | Number | % | Number | % | Number | % |
| Formal support | Employer | 40 | 14.7 | 21 | 15.3 | 144 | 29.2 |
| | Police | 149 | 54.6 | 9 | 6.6 | 148 | 30.0 |
| | Healthcare workers | 35 | 12.8 | 2 | 1.5 | 26 | 5.3 |
| | Civil society/ support | 108 | 39.6 | 10 | 7.3 | 22 | 4.5 |
| | Hotline | 15 | 5.5 | | – | | – |
| Informal support | Family | 72 | 26.4 | 72 | 52.6 | 116 | 23.5 |
| | Friends | 50 | 18.3 | 81 | 59.1 | 280 | 56.7 |
| | Colleagues | 31 | 11.4 | 31 | 22.6 | 50 | 10.1 |
| | Religious leader | 11 | 4.0 | 1 | 0.7 | 59 | 11.9 |
| | Social media | | – | 1 | 0.7 | 13 | 2.6 |
| | Miscellaneous | 7 | 2.6 | 6 | 4.4 | 5 | 1.0 |

Note: A participant might have more than one answer. The percentage was calculated separately in each row.

#273 participants responded "Yes" to the question "Do you know where to go for help when a migrant women worker experiences violence?"

^ 137 participants sought help when they experienced violence.

* Future plans for help-seeking were asked of all participants.

**Table 4. Ways of help received among women who sought help (n = 137).**

| Ways of help received | Frequency | % |
|---|---|---|
| Did not receive any help | 16 | 12.5 |
| Stepped in to halt the violence | 37 | 27.0 |
| Being asked preferred mode of assistance | 30 | 21.9 |
| Being given money | 25 | 18.3 |
| Provided a safe place for a rest/ living | 20 | 14.6 |
| Talked to the perpetrator who committed violence | 13 | 9.5 |
| Being taken for medical treatment | 7 | 5.1 |
| Gathered others for help | 6 | 4.4 |
| Informed the police | 3 | 2.2 |
| Informed the employer | 3 | 2.2 |
| Being taken to a civil society | 3 | 2.2 |
| Miscellaneous ways | 44 | 32.1 |

Note: A participant might have more than one answer. The percentage was calculated separately in each row based on the number of participants who sought help when they experienced violence = 137.

## Factors associated with lifetime IPV against women migrant workers

According to the multiple logistic regression analyses adjusted for age, education, and marital status (Table 6), the following factors show a statistically significant association with IPV against women migrant workers: area of residence in Thailand, duration of staying in Thailand, average income per month, Thai language proficiency, perceived safety in Thailand, and job loss during the COVID-19 pandemic ($p$–value <0.05 for all).

**Table 5. Impact of violence among survivors of violence (n = 269).**

| Impacts | | Number | % |
|---|---|---|---|
| Psychological impacts | Felt scared, insecure, or worried | 137 | 50.9 |
| | Felt angry | 113 | 42.0 |
| | Felt ashamed or embarrassed | 110 | 40.9 |
| | Felt sad, hopeless, or depressed | 75 | 27.9 |
| | Concerned about the safety of children | 27 | 10.0 |
| Physical impacts | Had a physical illness or injury | 64 | 23.8 |
| Impacts on job | Could not focus on work | 58 | 21.6 |
| | Being distracted while working | 41 | 15.2 |
| | Loss of confidence in work | 33 | 12.3 |
| | Needed to take a break from work | 27 | 10.0 |
| | Had to change job or employer | 26 | 9.7 |
| | Lost job | 24 | 8.9 |
| | Had to change departments or sections in the workplace | 2 | 0.7 |

Note: A participant might have more than one answer. The percentage was calculated separately for each row based on n = 269.

Compared to women who lived in Bangkok or its vicinity, those who lived in other provinces were 50% less likely to experience IPV (AOR = 0.50, 95% CI = 0.32, 0.81). Women who had lived five or more years in Thailand were two times more likely to have experienced IPV than those who had lived less than five years in Thailand (AOR = 2.18, 95% CI = 1.21, 3.93). Women with a higher average income were less likely to experience IPV than women with an income ≤ 5,000 THB per month (for monthly income 5,001–10,000 THB; AOR = 0.43, 95% CI = 0.23, 0.80 and for monthly income 10,001–15,000 THB; AOR = 0.51, 95% CI = 0.28, 0.94). Women who were able to speak and read Thai were 61% less likely to experience IPV than women who were not able to do so (AOR = 0.39, 95% CI = 0.17, 0.87). The perception of being unsafe in Thailand was significantly associated with women's experience of IPV in Thailand (AOR = 3.54, 95% CI = 2.18, 5.75). Women who lost their jobs during the COVID-19 pandemic were two times more likely to experience IPV than those who did not lose their jobs (AOR = 2.25, 95% CI = 1.41, 3.61).

### Factors associated with lifetime N-IPV against women migrant workers

According to the multiple logistic regression analyses adjusted for age, education, and marital status (Table 7), The statistically significant factors with N-IPV include area of residence in Thailand, duration of stay in Thailand, average income (THB per month), perceived safety in Thailand, job and income loss during the COVID-19 pandemic (p–value < 0.05 for all).

Compared to women who lived in Bangkok or its vicinity, those who lived in other provinces were 42% less likely to experience N-IPV (AOR = 0.58 (0.38, 0.88). Women who had lived five or more years in Thailand were two times more likely to experience N-IPV than those who lived less than five years in Thailand (AOR = 2.09, 95% CI = 1.28, 3.42). Women with an average income of 5,001–10,000 THB were 44% less likely to experience N-IPV than women with income ≤ 5,000 THB per month (AOR = 0.56, 95% CI = 0.32, 0.98). Women's perception that they were unsafe in Thailand was significantly associated with their experience of N-IPV in Thailand (AOR = 5.46, 95% CI = 3.53, 8.46). Women who lost their jobs or income during the COVID-19 pandemic were more likely to experience N-IPV than those who did not lose their jobs (AOR for job lost = 2.15, 95% CI = 1.41, 3.29, and AOR for income lost = 1.73, 95% CI = 1.15, 2.59).

**Table 6. Factors associated with lifetime IPV against women migrant workers (n = 405).**

| Characteristics of women migrant workers | IPV (%) | | Simple logistic regression | | Multiple logistic regression | |
|---|---|---|---|---|---|---|
| | No | Yes | COR (95% CI) | p-value [a] | AOR * (95% CI) | p-value [a] |
| Area of residence in Thailand | | | | | | |
| • Bangkok or its vicinity | 50.0 | 50.0 | Reference | | Reference | |
| • Other provinces | 69.2 | 39.8 | 0.45 (0.29, 0.68) | <0.001 | 0.50 (0.32, 0.81) | **0.004** |
| Duration of stay in Thailand | | | | | | |
| • Less than 5 years | 76.5 | 23.5 | Reference | | Reference | |
| • 5 years and above | 51.2 | 48.8 | 3.10 (1.86, 5.17) | **<0.001** | 2.18 (1.21, 3.93) | **0.009** |
| Average income (THB per month) | | | | | | |
| • Less than or equal to 5,000 | 48.2 | 51.8 | Reference | | Reference | |
| • 5,001–10,000 | 64.9 | 35.1 | 0.50 (0.29, 0.88) | **0.016** | 0.43 (0.23, 0.80) | **0.008** |
| • 10,001–15,000 | 62.9 | 37.1 | 0.55 (0.32, 0.94) | **0.030** | 0.51 (0.28, 0.94) | **0.031** |
| • 15,001 and above | 32.5 | 67.5 | 1.93 (0.88, 4.25) | 0.102 | 1.10 (0.46, 2.66) | 0.831 |
| Thai language proficiency | | | | | | |
| • Not able to speak or read | 45.5 | 54.5 | Reference | | Reference | |
| • Able to speak but not able to read | 51.7 | 48.3 | 0.78 (0.37, 1.64) | 0.512 | 0.72 (0.32, 1.62) | 0.429 |
| • Able to speak and read | 65.1 | 34.9 | 0.45 (0.21, 0.94) | **0.034** | 0.39 (0.17, 0.87) | **0.021** |
| Perceived safety in workplace, residence, travelling, or commuting in Thailand | | | | | | |
| • Safe | 73.6 | 26.4 | Reference | | Reference | |
| • Not safe | 42.3 | 57.7 | 3.80 (2.50, 5.78) | **<0.001** | 3.54 (2.18, 5.75) | **<0.001** |
| Lost job during the COVID-19 pandemic | | | | | | |
| • No | 64.6 | 35.4 | Reference | | Reference | |
| • Yes | 44.8 | 55.2 | 2.25 (1.48, 3.40) | **<0.001** | 2.25 (1.41, 3.61) | **0.001** |
| Lost income during the COVID-19 pandemic | | | | | | |
| • No | 64.0 | 36.0 | Reference | | Reference | |
| • Yes | 53.3 | 46.7 | 1.56 (1.04, 2.34) | **<0.034** | 1.29 (0.81, 2.05) | 0.279 |

Notes. COR = Crude odds ratio, AOR = Adjusted odds ratio, CI = Confidence interval, α = 0.05. a $p$–value < 0.05 are in bold.

* Age, education level, and marital status were adjusted.

## Discussion

This study investigated women migrant workers' experiences of violence and its associated factors across seven key provinces in Thailand during the COVID-19 pandemic. Moreover, we described impacts of violence on these women, women's help-seeking patterns and knowledge and future plans of help-seeking.

The results indicate that nearly half of the women in this study had experienced one or more forms of violence in their lifetime (42.47% for IPV and 46.15% for N-IPV). This prevalence is higher than that of violence against the general population of women in Myanmar, Laos, and Cambodia according to each country's nationwide surveys [25–27]; however, differences in methodology limit direct comparability. This higher prevalence is consistent with the fact that they are more vulnerable to violence because of their gender and migration status [7] and is aligned with a previous study that reported a high prevalence of violence against women migrant workers in Thailand [28]. Migration can be a challenging process, and many migrants experience various forms of violence at different stages of migration. In addition, we used a modified measurement tool to include experiences of violence such as economic abuse and cyberbullying. These items were not included in the original questionnaire developed by WHO [24], which might be a reason for the high prevalence of violence found in our study.

**Table 7. Factors associated with lifetime N-IPV against women migrant workers (n = 494).**

| Characteristics of women migrant workers | N-IPV (%) | | Simple logistic regression | | Multiple logistic regression | |
|---|---|---|---|---|---|---|
| | No | Yes | COR (95% CI) | p-value [a] | AOR * (95% CI) | p-value [a] |
| Area of residence in Thailand | | | | | | |
| • Bangkok or its vicinity | 48.1 | 51.9 | Reference | | Reference | |
| • Other provinces | 63.9 | 36.1 | 0.52 (0.36, 0.76) | 0.001 | 0.58 (0.38, 0.88) | **0.010** |
| Duration of stay in Thailand | | | | | | |
| • Less than 5 years | 70.3 | 29.7 | Reference | | Reference | |
| • 5 years and above | 46.8 | 53.2 | 2.68 (1.78, 4.05) | **<0.001** | 2.09 (1.28, 3.42) | **0.003** |
| Average income (THB per month) | | | | | | |
| • Less than or equal to 5,000 | 46.7 | 53.3 | Reference | | Reference | |
| • 5,001–10,000 | 61.0 | 39.0 | 0.56 (0.33, 0.94) | **0.029** | 0.56 (0.32, 0.98) | **0.043** |
| • 10,001–15,000 | 57.5 | 42.5 | 0.65 (0.39. 1.07) | 0.089 | 0.63 (0.37, 1.09) | 0.100 |
| • 15,001 and above | 27.7 | 72.3 | 2.29 (1.07, 4.90) | **0.033** | 1.60 (0.71, 3.63) | 0.260 |
| Thai language proficiency | | | | | | |
| • Not able to speak or read | 50.0 | 50.0 | Reference | | Reference | |
| • Able to speak but not able to read | 51.0 | 49.0 | 0.96 (0.48, 1.92) | 0.910 | 1.02 (0.49, 2.12) | 0.953 |
| • Able to speak and read | 56.6 | 43.4 | 0.77 (0.39, 1.51) | 0.443 | 0.82 (0.40, 1.68) | 0.593 |
| Perceived safety in workplace, residence, travelling and commuting in Thailand | | | | | | |
| • Safe | 74.1 | 25.9 | Reference | | Reference | |
| • Not safe | 33.6 | 66.4 | 5.65 (3.83, 8.33) | **<0.001** | 5.46 (3.53, 8.46) | **<0.001** |
| Lost job during the COVID-19 pandemic | | | | | | |
| • No | 60.4 | 39.6 | Reference | | Reference | |
| • Yes | 41.0 | 59.0 | 2.20 (1.50, 3.21) | **<0.001** | 2.15 (1.41, 3.29) | **<0.001** |
| Lost income during the COVID-19 pandemic | | | | | | |
| • No | 63.1 | 36.9 | Reference | | Reference | |
| • Yes | 47.2 | 52.8 | 1.91 (1.33, 2.76) | **0.001** | 1.73 (1.15, 2.59) | **0.008** |

Notes. COR = Crude odds ratio, AOR = Adjusted odds ratio, CI = Confidence interval, α = 0.05. a p–value < 0.05 are in bold.

* Age, education level, and marital status were adjusted.

In this study, nearly half of the women did not have knowledge of where to seek help if a migrant woman experienced violence. This finding emphasizes the need to promote knowledge and awareness of violence, including how to contact social, legal, and health support services. Confirming the findings from a previous study in Thailand [4], nearly half of the women in this study who had experienced violence (50.9%) did not seek help. This result is lower than that of a study in Uganda (63.5%) but higher than findings in other developing countries [29,30]. Additionally, only 6.6% of women who sought help reported to the police although 54.6% of women who knew where to seek help for violence answered that they could report it to the police. In addition, less than one-third of women (30.0%) would request help from the police as part of a future plan of help-seeking for violence. The findings show that having knowledge does not necessarily lead to actual behaviour.

These results indicate that there may be barriers to reporting to the police, such as fear of retaliation, language barriers, or little trust in the justice system. The justice system may not always be culturally sensitive or attuned to the specific needs of migrant women. Differences in cultural norms and expectations might create a reluctance to engage with law enforcement. Migrant women may fear judgment or cultural insensitivity, impacting their willingness to seek help from the police. To address these barriers, it is crucial to implement targeted interventions and support services that consider the unique challenges faced by migrant women when seeking help after experiencing violence. These interventions should

focus on building trust in the justice system, providing culturally sensitive assistance, offering language support, and raising awareness about available support services. Additionally, efforts should be made to create an environment in which migrant women feel safe and empowered to come forward without fear of retaliation or negative consequences.

Concerning the types of help received, 12.5% of women survivors did not receive any help even when they sought it. The possible reasons for this might be related to the absence of protection orders, limited referral networks and coordination mechanisms, and negative public attitudes towards migrant workers in Thailand [4]. Regarding the impact of violence on survivors, more than half of women suffered from one or more forms of psychological impact. In addition, nearly one-fourth of women suffered from physical illness or injury, and more than one-fifth of women were found to have experienced negative impacts at their jobs. To reduce the impacts of violence on survivors, it is critical to improve access to comprehensive services, including psychosocial support, economic empowerment programs, and job opportunities [31]. Furthermore, encouraging a supportive environment and providing resources to help survivors cope with the effects of violence, such as psychological counselling and legal assistance, could contribute to mitigating the negative impacts of violence on women.

Regarding the factors associated with violence against women migrant workers, multivariable analyses identified that women residing in provinces other than Bangkok were associated with lower odds of reporting both IPV and N-IPV; this could be attributed to several factors. Women living in provinces other than Bangkok may benefit from more tightly knit social circles, fostering a sense of community support and protection. In contrast to the potentially isolated urban environment of Bangkok, other provinces may offer a more cohesive community structure, in which individuals are interconnected and responsive to each other's well-being. This social cohesion might result in easier access to support services and protection mechanisms, acting as a deterrent against both IPV and N-IPV. Additionally, the dynamics of urban living in Bangkok may contribute to increased anonymity and reduced community surveillance, potentially making it more challenging for women to seek help or for support networks to intervene in cases of violence. These findings underscore the importance of considering the contextual nuances of different geographic locations in addressing and preventing violence against women migrant workers.

In line with previous studies and researchers' expectations, women with a higher income in this study were associated with lower odds of reporting violence experiences, as greater economic independence and the empowerment of women are protective against violence [31,32]. Poverty significantly affects women, trapping them in abusive situations and limiting their ability to escape [33]. These findings are consistent with a broader understanding that poverty can exacerbate vulnerabilities. The link between higher income and lower violence suggests that economic empowerment programs and initiatives aimed at financially empowering women may be effective preventive measures against violence against them. Furthermore, addressing the economic disparities that contribute to gender-based violence is critical to establishing a more equitable and just society. This highlights the importance of multifaceted interventions that address not only the immediate consequences of violence but also the underlying socioeconomic factors that perpetuate such abuse.

In this study, women with a longer duration of stay in Thailand and women who perceived themselves to be unsafe in Thailand were associated with an increased likelihood of IPV and N-IPV. This suggests several potential dynamics. In our study, we observed that longer-term residents might encounter challenges unique to their extended exposure to the host culture. Over time, they may face increased expectations and pressures to conform, leading to heightened stress and vulnerability. Additionally, as cultural dynamics evolve, long-term residents may find it challenging to keep pace with changes, contributing to their difficulties in navigating the local environment. We acknowledge that acculturation is a multifaceted process influenced by various factors, and our findings suggest that a longer duration of stay may not always correlate with a smoother acculturation experience. Furthermore, women who perceive themselves as unsafe may be more vulnerable due to factors such as discrimination, a lack of social integration, or a sense of insecurity in their surroundings.

Interestingly, women who lost their jobs during the COVID-19 pandemic were associated with higher likelihood of reporting IPV and N-IPV experiences. Studies have shown that the frequency and severity of violence against women

migrant workers have risen due to lockdowns and other pandemic-related factors, leading to an intensified risk of violence for those who have lost their jobs [34,35]. The economic impact of job loss and the difficulty of returning to their home countries due to travel restrictions also made migrant women more vulnerable to abuse and exploitation [7,36].

This study achieved a comprehensive understanding of violence against women migrant workers from Cambodia, Lao PDR, and Myanmar during the COVID-19 pandemic. However, the findings may not be broadly applicable, as data were collected in specific provinces. The use of snowball sampling introduced potential selection bias, so the sample may not be representative and generalizable. There is a possibility that women with experiences of violence were more likely to participate in the study, thereby influencing the sample. Given the reliance on social networks, participants with prior experiences of violence or stronger community connections may have been more likely to be recruited, potentially leading to an overrepresentation of survivors and inflation of prevalence estimates. Other limitations include the use of enumerators in interviews, self-reporting bias, and recall bias. To mitigate these challenges, the study implemented comprehensive training for enumerators, closely monitored the data collection processes, and collaborated with local non-governmental organizations (NGOs) and civil society organizations (CSOs). These measures were crucial not only for minimizing biases but also for maintaining participant privacy and establishing rapport during the data collection process.

## Recommendations and conclusion

Our study identified a high prevalence of multiple forms of violence against women migrant workers, highlighting an urgent need for targeted, survivor-centred actions that address key factors identified in this study, including area of residence, duration of stay, monthly income, perceived safety, Thai language proficiency, and job or income loss during the COVID-19 pandemic. Policymakers and service providers should ensure that support services are accessible, responsive, and specifically tailored to the unique needs of women migrant workers, including strengthening multilingual and migrant-sensitive services within existing mechanisms such as workplace-based protections, labour inspection systems, Migrant Worker Assistance Centres, and health-sector One-Stop Crisis Centres in Thailand. Expanding Thai language programs and communication support is essential to improve migrant women's ability to understand their rights, report violence, and access appropriate formal support. In addition, strengthening peer networks among migrant women can foster a supportive environment where experiences are shared, and collective resilience is built, complementing formal support systems that are currently underutilized. Moreover, comprehensive, culturally relevant educational campaigns should be implemented, incorporating feedback from women migrant workers to enhance their effectiveness, and aligned with national gender equality, anti-trafficking, and business and human rights frameworks to promote violence prevention in migrant communities and workplaces. Furthermore, ongoing research is essential to monitor the issue and inform future responses, supporting evidence-based implementation and evaluation of national policies addressing migrant women's welfare.

In summary, our findings underscore the imperative need to address violence against women migrant workers who have been facing intersecting forms of vulnerability and marginalization. Comprehensive interventions are recommended to prevent violence and respond effectively at both individual and community levels, supporting survivors and addressing associated risk factors, while reinforcing Thailand's existing legal and policy commitments to violence prevention, gender equality, and migrant worker protection. Importantly, a collaborative, multifaceted approach among policymakers and stakeholders is crucial for creating a safer, more supportive environment, ensuring that national policies are translated into practical, survivor-centred actions that contribute to the elimination of violence against this marginalized population.

## Supporting information

**S1 File. Quantitative dataset of this study.**
(XLSX)

## Author contributions

**Conceptualization:** Montakarn Chuemchit, Ratana Somrongthong.

**Data curation:** Montakarn Chuemchit, Nyan Linn, Suttharuethai Chernkwanma.

**Formal analysis:** Montakarn Chuemchit, Nyan Linn, Chit Pyae Pyae Han, Zayar Lynn, Nutta Taneepanichskul, Wandee Sirichokchatchawan.

**Funding acquisition:** Montakarn Chuemchit.

**Investigation:** Montakarn Chuemchit.

**Methodology:** Montakarn Chuemchit, Nyan Linn, Suttharuethai Chernkwanma, Nutta Taneepanichskul, Wandee Sirichokchatchawan.

**Project administration:** Suttharuethai Chernkwanma.

**Supervision:** Montakarn Chuemchit, Ratana Somrongthong.

**Validation:** Montakarn Chuemchit.

**Visualization:** Montakarn Chuemchit, Nyan Linn, Chit Pyae Pyae Han, Zayar Lynn, Nutta Taneepanichskul, Wandee Sirichokchatchawan.

**Writing – original draft:** Montakarn Chuemchit, Nyan Linn.

**Writing – review & editing:** Montakarn Chuemchit, Nyan Linn.

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
