## [Decision Letter · Decision Letter 0]

18 Dec 2025

Dear Dr. Chuemchit,

Thank you for submitting your manuscript to PLOS ONE. After careful consideration, we feel that it has merit but does not fully meet PLOS ONE’s publication criteria as it currently stands. Therefore, we invite you to submit a revised version of the manuscript that addresses the points raised during the review process.

We look forward to receiving your revised manuscript.

Kind regards,

Tatchalerm Sudhipongpracha

Academic Editor

PLOS One

**Journal Requirements:**

2. We note that only one reference number for your ethics documents is listed in the ethics section of your manuscript. Please add the original ethics document reference number (COA.No.232/2020) to your manuscript.

“This study was funded by the EU funded Spotlight initiative, Safe and Fair Program, UN Women Thailand (PSA-ROAP-2021-008).”

5. For studies involving third-party data, we encourage authors to share any data specific to their analyses that they can legally distribute. PLOS recognizes, however, that authors may be using third-party data they do not have the rights to share. When third-party data cannot be publicly shared, authors must provide all information necessary for interested researchers to apply to gain access to the data. (https://journals.plos.org/plosone/s/data-availability#loc-acceptable-data-access-restrictions)

Reviewers' comments:

Reviewer's Responses to Questions

**Comments to the Author**

1. Is the manuscript technically sound, and do the data support the conclusions?

Reviewer #1: Yes

Reviewer #2: Yes

2. Has the statistical analysis been performed appropriately and rigorously?

Reviewer #1: Yes

Reviewer #2: Yes

3. Have the authors made all data underlying the findings in their manuscript fully available?

Reviewer #1: No

Reviewer #2: No

4. Is the manuscript presented in an intelligible fashion and written in standard English?

Reviewer #1: Yes

Reviewer #2: Yes

Reviewer #1: The manuscript titled “Violence Against Women Migrant Workers in Thailand: A Cross-Sectional Study on Experiences, Impact, and Help Seeking” presents an important and timely contribution to the field of public health and gender-based violence research. The topic is highly relevant, particularly in the Southeast Asian context where migrant women face intersecting vulnerabilities due to gender, socioeconomic status, and migration status. The study is ethically sound, methodologically rigorous, and presented in a clear and structured manner. Below are detailed comments and suggestions:

1. Technical Soundness and Data Support for Conclusions

The research is methodologically solid. The use of a cross-sectional design is appropriate for assessing the prevalence and correlates of violence among women migrant workers. The sample size (n = 494) across seven provinces provides substantial breadth for descriptive and inferential analysis.

The study instruments are well-grounded in existing validated tools, particularly the WHO multi-country study questionnaire, which enhances validity. Internal reliability is strong (Cronbach’s α = 0.8–0.9).

The conclusions are appropriately drawn and align with the results. The authors correctly refrain from making causal claims and discuss limitations related to the sampling strategy and potential biases. The policy implications and recommendations are relevant and well-articulated.

Suggestion: Consider briefly emphasizing how findings can inform specific interventions or existing national policies addressing migrant women’s welfare in Thailand to enhance applied significance.

2. Statistical Analysis

The statistical analyses are appropriate and performed with good rigor. Descriptive statistics and logistic regression analyses (both crude and adjusted models) are used effectively to explore associations between demographic and contextual variables with IPV and N-IPV. Adjustments for potential confounders (age, education, marital status) are suitable.

The tables are clear, and significant results are correctly interpreted in the discussion.

Minor recommendations:

• State whether multicollinearity among predictors was tested and ruled out.

These additions would further strengthen transparency and confidence in the analytical rigor.

3. Data Availability

The Data Availability Statement is transparent and ethically justified. Quantitative data are reported to be fully available within the manuscript, while qualitative data are restricted due to the sensitive nature of violence narratives. This is an acceptable limitation under ethical standards; however, it does not fully comply with PLOS ONE’s open data policy. However, the restricted access to qualitative data is understandable, provided that ethical constraints are clearly documented, as already done.

4. Ethical Considerations

The study adheres to appropriate ethical standards, with approval from Chulalongkorn University Ethics Review Committee (COA.No.196/2021). Informed consent procedures were well described, and participants’ confidentiality was prioritized.

The study deals with a highly sensitive topic and demonstrates commendable ethical awareness, especially in using trained female enumerators from migrant communities and providing referral information to participants after interviews.

Overall Assessment

This manuscript makes a valuable empirical and policy-relevant contribution to the understanding of violence against women migrant workers in Thailand. The design, analysis, and interpretation are appropriate, and the findings are clearly presented and well-supported by the data.

Strengths:

• Robust methodology and sample size

• Ethical and culturally sensitive data collection

• Clear presentation and actionable recommendations

Areas for minor improvement:

• State whether multicollinearity among predictors was tested and ruled out

• Consider briefly highlighting how the findings can inform specific interventions or align with existing national policies

Reviewer #2: Recommendation:

The manuscript presents original and policy-relevant empirical findings and predominantly satisfies PLOS ONE’s scientific standards (robust analyses, ethical approval, and clarity of presentation). But before the paper can be accepted, some important problems need to be fixed: (1) inconsistent or unclear information about ethics approval, (2) incomplete or inconsistent information about data availability that doesn't follow PLOS policy, (3) not enough methodological detail in some areas (sampling, questionnaire validation, handling missing data, power/sample-size justification, and selection of covariates), and (4) clarification about how the findings overlap with those that have already been published. Below, I list the strengths, concerns (with proof from the file that was sent in), and specific changes that need to be made.

Strengths

• The subject matter is significant and insufficiently researched; violence against female migrant workers in Thailand is critically pertinent to policy, and the manuscript contributes empirical data from various provinces.

• The sample size is sufficiently large (n = 494) for cross-sectional prevalence and logistic regression analyses.

• The questionnaire seems to be based on WHO tools and has extra areas (economic abuse, cyberbullying) with an internal reliability score of about 0.8–0.9, which makes the measurement more credible.

• Analyses employ logistic regression, yielding both crude and adjusted odds ratios (AOR); the models account for age, education, and marital status, and display confidence intervals (CIs) and p-values. There are tables of AOR/COR that are easy to read and understand.

Major Concerns

1. Ethics approval information that isn't always clear (needs to be fixed and made clear).

• There are two different COA numbers in the submission: COA.No.232/2020 and COA.No.196/2021. These numbers are shown in different parts of the file. The Methods section says that data collection was approved (COA.No.196/2021), but other administrative fields say COA.No.232/2020. Please clarify which protocol(s) approved this study, the name of the committee that did so, the approval number(s) and date(s), and whether any changes or secondary analyses were included in the approval. If there are more than one approval, please explain clearly why (for example, if there are different substudies).

2. The data availability statement does not follow PLOS ONE's rules as they are now written or make inconsistent claims.

• The manuscript makes the claim "All relevant data are within the manuscript and its Supporting Information files" in several places. It also says that "datasets used and analyzed are available from the corresponding author upon reasonable request," but qualitative transcripts are not available because they are sensitive and are held by a third party through UN Women. PLOS ONE generally wants the quantitative data that supports the findings to be publicly accessible in a repository or stored with the right protections. "Available on request" is usually not enough. Please:

o Please say clearly which quantitative dataset(s) (like an anonymized survey CSV) will be deposited and give the repository name and accession/DOI (or say why deposition is not possible).

o If you really can't share qualitative data, be clear about the exact limits and give people a way to get to it in a controlled way (for example, by giving them de-identified excerpts, a contact for the institutional data access committee, or a URL/email for the data custodian). The current contact information for UN Women is helpful, but it doesn't include a full compliance plan.

3. There is some overlap with previous publications, so it needs to be made clear that it is new and not a duplicate.

• The manuscript states, “partial findings were previously published (Chuemchit et al., 2024) focusing on discrimination….” (Lines 127-128). The authors must explicitly delineate (a) the analyses/results/figures/tables that are novel in this submission, (b) any content that overlaps with the preceding paper, and (c) the rationale for why the current paper does not represent redundant publication. Give citations and, if you can, a short table of overlap (for example, "Table X in previous paper = data on discrimination; this paper uses the same sample but presents new analyses on lifetime IPV/N-IPV and help-seeking"). The journal policy says this is necessary.

4. Sampling and generalizability: the limitations of snowball sampling require more thorough examination and the quantification of potential bias.

• Snowball sampling is suitable for elusive populations but is subject to selection bias. The manuscript recognizes its limitations; however, it should also delineate recruitment chains, seed selection criteria, the number of waves employed, any efforts made to diversify seeds, and offer a critical analysis of how these factors may skew prevalence estimates (e.g., through oversampling survivors). If possible, show sensitivity analyses or compare the demographics of the sample to external benchmarks (like national or regional distributions of migrant workers) to see how representative it is.

5. Statistical methods: need more detail and checks for strength.

• The main methods used were logistic regressions and taking into account age, education, and marital status. Please give:

o Why we chose these covariates instead of others, like income or length of stay, which are shown in tables as exposures.

o Handling of missing data (how much data is missing for each variable and whether listwise deletion, imputation, or other methods were used).

o If the correction for multiple comparisons was taken into account (a lot of associations were tested).

o If you can, put unadjusted and fully adjusted model tables in the main text or the supplement.

6. There is no sample size or power calculation.

• The paper must indicate whether a priori sample size or power calculations were conducted to substantiate the use of n=494 for the primary outcomes (prevalence estimates, detection of associations). If none was done, add a short discussion of detectable effect sizes based on the sample.

7. Measurement validation: additional information required regarding questionnaire validation and reliability.

• The manuscript states that three experts verified content validity and presents reliability coefficients (α = 0.8, 0.9). Please include information on how experts were chosen, whether cognitive interviewing or pilot testing was done, how translation and back-translation were done, and item-level psychometrics (if available). This is significant as the instrument was adapted from WHO tools to incorporate new domains (economic/cyber violence).

8. Interpretation of the findings.

• The Discussion section presently contains multiple assertions that suggest causality (e.g., “X leads to Y,” “Factor A causes higher rates of violence,” “B results in increased risk of C”). Because this is a cross-sectional study, the data cannot be used to draw conclusions about cause and effect. Associations can be reported, but establishing cause-and-effect relationships necessitates longitudinal or experimental designs.

• The authors need to change the language in the Discussion to show that the findings are related. For instance, phrases like "X is linked to Y" or "people who said X were more likely to say Y" would be better than causal statements.

• The study makes a number of cross-country comparisons, which suggest that the patterns or prevalence found in this study are "higher" or "lower" than those found in other contexts. It is important to be careful when making these comparative claims because the sampling frames, measurement tools, and study designs are all very different. For instance, it would be more accurate to say, "The prevalence observed in this non-probability sample of migrant women appears higher than figures reported in national surveys from country Z; however, differences in methodology limit direct comparability."

**Do you want your identity to be public for this peer review?** For information about this choice, including consent withdrawal, please see our For information about this choice, including consent withdrawal, please see our Privacy Policy .

Reviewer #1: **Yes:** Hsu Nandar AungHsu Nandar Aung

Reviewer #2: **Yes:** Kosum OmphornuwatKosum Omphornuwat

---

## [Author Response · Author response to Decision Letter 1]

29 Jan 2026

Comments to the Author

1. Is the manuscript technically sound, and do the data support the conclusions?

- Reviewer #1: Yes

- Reviewer #2: Yes

2. Has the statistical analysis been performed appropriately and rigorously?

- Reviewer #1: Yes

- Reviewer #2: Yes

3. Have the authors made all data underlying the findings in their manuscript fully available? In addition to summary statistics, the data points behind means, medians and variance measures should be available. If there are restrictions on publicly sharing data—e.g. participant privacy or use of data from a third party—those must be specified.

- Reviewer #1: No

- Reviewer #2: No

Thank you for your comments. The authors have revised the “Data availability statement” in the manuscript as “All relevant data are within the paper. Quantitative data is openly accessible.” We have uploaded the dataset to the system. The qualitative data are totally not presented/ described/ used/ analysed in this paper. Therefore, the qualitative data is not relevant to mention for this current paper.

Comments from reviewers

Reviewer #1:

The manuscript titled “Violence Against Women Migrant Workers in Thailand: A Cross-Sectional Study on Experiences, Impact, and Help Seeking” presents an important and timely contribution to the field of public health and gender-based violence research. The topic is highly relevant, particularly in the Southeast Asian context where migrant women face intersecting vulnerabilities due to gender, socioeconomic status, and migration status. The study is ethically sound, methodologically rigorous, and presented in a clear and structured manner. Below are detailed comments and suggestions:

Thank you for your appreciation and support for our manuscript. The authors really appreciate your comments and suggestions.

1. Technical Soundness and Data Support for Conclusions

The research is methodologically solid. The use of a cross-sectional design is appropriate for assessing the prevalence and correlates of violence among women migrant workers. The sample size (n = 494) across seven provinces provides substantial breadth for descriptive and inferential analysis. The study instruments are well-grounded in existing validated tools, particularly the WHO multi-country study questionnaire, which enhances validity. Internal reliability is strong (Cronbach’s α = 0.8–0.9). The conclusions are appropriately drawn and align with the results. The authors correctly refrain from making causal claims and discuss limitations related to the sampling strategy and potential biases. The policy implications and recommendations are relevant and well-articulated.

Suggestion: Consider briefly emphasizing how findings can inform specific interventions or existing national policies addressing migrant women’s welfare in Thailand to enhance applied significance.

Thank you for your comments. The authors have revised the manuscript according to the reviewer’s suggestions as follows.

Conclusion of the Abstract (line 53 – 59): “Key strategies include strengthening workplace-based protections, multilingual reporting mechanisms, peer networks, culturally relevant education, and individual- and community-level interventions within labour, health, and social protection systems. Collaborative efforts among policymakers and stakeholders are vital for the prevention of and response to violence and for ensuring the safety of women migrant workers.”

Recommendations and conclusions (Line 472 – 503): “Our study identified a high prevalence of multiple forms of violence against women migrant workers, highlighting an urgent need for targeted, survivor-centred actions that address key risk factors identified in this study, including area of residence, duration of stay, monthly income, perceived safety, Thai language proficiency, and job or income loss during the COVID-19 pandemic. Policymakers and service providers ..., including strengthening multilingual and migrant-sensitive services within existing mechanisms such as workplace-based protections, labour inspection systems, Migrant Worker Assistance Centres, and health-sector One-Stop Crisis Centres in Thailand. Expanding Thai language programs and communication support is essential to improve migrant women’s ability to understand their rights, report violence, and access appropriate formal support. In addition, ... complementing formal support systems that are currently underutilized. ... and aligned with national gender equality, anti-trafficking, and business and human rights frameworks to promote violence prevention in migrant communities and workplaces. Furthermore, ... supporting evidence-based implementation and evaluation of national policies addressing migrant women’s welfare. ... In summary, ... while reinforcing Thailand’s existing legal and policy commitments to violence prevention, gender equality, and migrant worker protection. Importantly, ... ensuring that national policies are translated into practical, survivor-centred actions that contribute to the elimination of violence against this marginalized population.”

2. Statistical Analysis

The statistical analyses are appropriate and performed with good rigor. Descriptive statistics and logistic regression analyses (both crude and adjusted models) are used effectively to explore associations between demographic and contextual variables with IPV and N-IPV. Adjustments for potential confounders (age, education, marital status) are suitable.

The tables are clear, and significant results are correctly interpreted in the discussion.

Minor recommendations: State whether multicollinearity among predictors was tested and ruled out. These additions would further strengthen transparency and confidence in the analytical rigor.

Thank you for your comments. The authors have revised the manuscript according to the reviewer’s suggestions as follows.

Line 248 – 249: “Multicollinearity among predictors was tested; Variance Inflation Factor (VIF) of all variables were ≤ 5, indicating no issue of multicollinearity.”

3. Data Availability

The Data Availability Statement is transparent and ethically justified. Quantitative data are reported to be fully available within the manuscript, while qualitative data are restricted due to the sensitive nature of violence narratives. This is an acceptable limitation under ethical standards; however, it does not fully comply with PLOS ONE’s open data policy. However, the restricted access to qualitative data is understandable, provided that ethical constraints are clearly documented, as already done.

Thank you for your comments. The authors have revised the “Data availability statement” in the manuscript as “All relevant data are within the paper. Quantitative data is openly accessible.” We have uploaded the dataset to the system. The qualitative data are not presented/ described/used/ analysed in this paper. Therefore, the qualitative data is not relevant to mention for this current paper.

4. Ethical Considerations

The study adheres to appropriate ethical standards, with approval from Chulalongkorn University Ethics Review Committee (COA.No.196/2021). Informed consent procedures were well described, and participants’ confidentiality was prioritized. The study deals with a highly sensitive topic and demonstrates commendable ethical awareness, especially in using trained female enumerators from migrant communities and providing referral information to participants after interviews.

The authors express sincere thanks to the reviewer. The original ethics approval for this study was granted by the Chulalongkorn University Ethics Review Committee (COA.No.232/2020). The approval number COA.No.196/2021 refers to the approved extension. We have revised the manuscript to clearly report both ethics reference numbers.

Overall Assessment

This manuscript makes a valuable empirical and policy-relevant contribution to the understanding of violence against women migrant workers in Thailand. The design, analysis, and interpretation are appropriate, and the findings are clearly presented and well-supported by the data.

Strengths:

• Robust methodology and sample size

• Ethical and culturally sensitive data collection

• Clear presentation and actionable recommendations

Areas for minor improvement:

• State whether multicollinearity among predictors was tested and ruled out

• Consider briefly highlighting how the findings can inform specific interventions or align with existing national policies

Thank you for your comments and suggestions. The authors have revised the manuscript according to the reviewer’s recommendations as mentioned above.

Reviewer #2

The manuscript presents original and policy-relevant empirical findings and predominantly satisfies PLOS ONE’s scientific standards (robust analyses, ethical approval, and clarity of presentation). However, before the paper can be accepted, some important problems need to be fixed: (1) inconsistent or unclear information about ethics approval, (2) incomplete or inconsistent information about data availability that does not follow PLOS policy, (3) insufficient methodological detail in some areas (sampling, questionnaire validation, handling missing data, power/sample-size justification, and selection of covariates), and (4) clarification regarding how the findings overlap with those that have already been published.

Below, I list the strengths, concerns (with proof from the file that was sent in), and specific changes that need to be made.

The authors appreciate the reviewer’s comments and suggestions. We have made the required revision to the manuscript as detailed below.

Strengths

• The subject matter is significant and insufficiently researched; violence against female migrant workers in Thailand is critically pertinent to policy, and the manuscript contributes empirical data from various provinces.

• The sample size is sufficiently large (n = 494) for cross-sectional prevalence and logistic regression analyses.

• The questionnaire appears to be based on WHO tools and includes additional domains (economic abuse, cyberbullying), with internal reliability scores of approximately 0.8–0.9, which strengthens measurement credibility.

• Analyses employ logistic regression, yielding both crude and adjusted odds ratios (AOR). The models account for age, education, and marital status, and report confidence intervals (CIs) and p-values. The AOR/COR tables are clear and easy to interpret.

The authors appreciate the reviewer’s comments and suggestions. We have made the required revision to the manuscript as detailed below.

Major Concerns

1. Ethics approval information is unclear and inconsistent (must be clarified).

• Two different COA numbers are reported in the submission: COA.No.232/2020 and COA.No.196/2021. These numbers appear in different sections of the file. The Methods section states that data collection was approved under COA.No.196/2021, while other administrative fields report COA.No.232/2020.

Please clarify which protocol(s) approved this study, the name of the approving committee, the approval number(s) and date(s), and whether any amendments or secondary analyses were included in the approval. If more than one approval was obtained, please clearly explain why (for example, if different sub-studies were involved).

Thank you for your comments. This study was approved by the Chulalongkorn University Ethics Review Committee (COA.No.232/2020), Thailand, on 8 October 2020, with an approved extension under COA.No.196/2021 on 8 October 2021. We have revised the manuscript to clearly report both ethics reference numbers. There were no amendments and no secondary analyses made in this current paper as well as in the original research project.

2. The data availability statement does not comply with current PLOS ONE policy and contains inconsistencies.

• The manuscript repeatedly states that “All relevant data are within the manuscript and its Supporting Information files.” It also claims that “datasets used and analysed are available from the corresponding author upon reasonable request,” while noting that qualitative transcripts are unavailable because they are sensitive and held by a third party (UN Women).

PLOS ONE generally requires that quantitative data supporting the findings be publicly accessible via a repository or shared under appropriate protections. “Available upon request” is typically insufficient. Please address the following:

o Clearly specify which quantitative dataset(s) (e.g., an anonymized survey CSV) will be deposited, including the repository name and accession number/DOI, or explicitly explain why deposition is not possible.

o If qualitative data cannot be shared, clearly state the limitations and provide a controlled access mechanism (e.g., de-identified excerpts, contact details for an institutional data access committee, or a URL/email for the data custodian). While the current UN Women contact information is useful, it does not yet constitute a full compliance plan.

Thank you for your comments. The authors have revised the “Data availability statement” in the manuscript as “All relevant data are within the paper. Quantitative data is openly accessible.” We have uploaded the dataset to the system. The qualitative data are not presented/ described/used/ analysed in this paper. Therefore, the qualitative data is not relevant to mention for this current paper.

3. Overlap with previous publications must be clearly delineated to avoid redundancy.

• The manuscript states that “partial findings were previously published (Chuemchit et al., 2024) focusing on discrimination…” (Lines 127–128).

The authors must explicitly describe:

(a) which analyses, results, figures, and tables are novel in this submission;

(b) which elements overlap with the prior publication; and

(c) why the current manuscript does not constitute redundant publication.

Please include appropriate citations and, if possible, a brief table summarizing overlap (e.g., “Table X in the previous paper reports discrimination data; the current paper uses the same sample but presents new analyses on lifetime IPV/N-IPV and help-seeking”). This is required under journal policy.

Thank you for this important comment. The authors have put the required information in the revised manuscript according to your suggestions as below.

Materials and methods line 137 – 145. “Although this study uses the same sample from the research project and therefore shares general characteristics with the preceding paper, the analyses and results presented in the current paper, including all tables and figures, are entirely novel. This paper presents new analyses and critical findings on women’s lifetime experiences of IPV and non-IPV, their impacts, associated factors, and patterns of help-seeking, including ways help was received, knowledge of help-seeking, and future help-seeking plans. Accordingly, the current paper addresses separate analytical aims and contributes to original evidence that is not reported in the preceding publication and therefore does not constitute redundant publication. Findings of this paper provide a significant contribution to understanding the multifaceted challenges faced by migrant women workers.”

4. Sampling and generalizability require deeper examination and bias assessment.

• Snowball sampling is appropriate for hard-to-reach populations but is subject to selection bias. While the manuscript acknowledges this limitation, it should also describe recruitment chains, seed selection criteria, the number of recruitment waves, and any steps taken to diversify seeds.

A more critical discussion is needed regarding how these factors may bias prevalence estimates (e.g., oversampling survivors). If feasible, include sensitivity analyses or comparisons between sample demographics and external benchmarks (such as national or regional migrant worker statistics) to assess representativeness.

The authors thank to the reviewer for these important suggestions. Accordingly, revision was done in Study design and sampling techniques Line 166 – 176 and in discussion (limitation Line 463 – 465).

Seed selection criteria

“Initial seeds were purposively selected through community-based organizations and outreach workers based on eli

---

## [Decision Letter · Decision Letter 1]

10 Mar 2026

Violence Against Women Migrant Workers in Thailand: A Cross-Sectional Study on Experiences, Impact, and Help Seeking

PONE-D-25-37884R1

Dear Dr. Chuemchit,

We’re pleased to inform you that your manuscript has been judged scientifically suitable for publication and will be formally accepted for publication once it meets all outstanding technical requirements.

Kind regards,

Tatchalerm Sudhipongpracha

Academic Editor

PLOS One

Additional Editor Comments (optional):

Reviewers' comments:

Reviewer's Responses to Questions

**Comments to the Author**

Reviewer #2: All comments have been addressed

2. Is the manuscript technically sound, and do the data support the conclusions?

Reviewer #2: Yes

3. Has the statistical analysis been performed appropriately and rigorously?

Reviewer #2: Yes

4. Have the authors made all data underlying the findings in their manuscript fully available?

Reviewer #2: Yes

5. Is the manuscript presented in an intelligible fashion and written in standard English?

Reviewer #2: Yes

Reviewer #2: (No Response)

**Do you want your identity to be public for this peer review?** For information about this choice, including consent withdrawal, please see our For information about this choice, including consent withdrawal, please see our Privacy Policy .

Reviewer #2: No

---

## [Editor Report · Acceptance letter]

PONE-D-25-37884R1

PLOS One

Dear Dr. Chuemchit,

I'm pleased to inform you that your manuscript has been deemed suitable for publication in PLOS One. Congratulations! Your manuscript is now being handed over to our production team.

Kind regards,

on behalf of

Dr. Tatchalerm Sudhipongpracha

Academic Editor

PLOS One